# Radiotracers for Imaging of Inflammatory Biomarkers TSPO and COX-2 in the Brain and in the Periphery

**DOI:** 10.3390/ijms242417419

**Published:** 2023-12-13

**Authors:** Bright Chukwunwike Uzuegbunam, Christoph Rummel, Damiano Librizzi, Carsten Culmsee, Behrooz Hooshyar Yousefi

**Affiliations:** 1Nuclear Medicine Department, Klinikum Rechts der Isar, Technical University of Munich, 81675 Munich, Germany; b.uzuegbunam@tum.de; 2Institute of Veterinary Physiology and Biochemistry, Justus Liebig University Giessen, 35392 Gießen, Germany; christoph.d.rummel@vetmed.uni-giessen.de; 3Center for Mind Brain and Behavior, Universities Giessen and Marburg, 35043 Marburg, Germany; culmsee@uni-marburg.de; 4Department of Nuclear Medicine, Philipps University of Marburg, 35043 Marburg, Germany; librizzi@med.uni-marburg.de; 5Institute of Pharmacology and Clinical Pharmacy, Philipps University of Marburg, 35037 Marburg, Germany

**Keywords:** tracer development, neuroinflammation, pneumonia, myocarditis, translocator protein (TSPO), cyclooxygenase-2 (COX-2), positron emission tomography (PET), single-photon emission tomography (SPECT)

## Abstract

Inflammation involves the activation of innate immune cells and is believed to play an important role in the development and progression of both infectious and non-infectious diseases such as neurodegeneration, autoimmune diseases, pulmonary and cancer. Inflammation in the brain is marked by the upregulation of translocator protein (TSPO) in microglia. High TSPO levels are also found, for example, in macrophages in cases of rheumatoid arthritis and in malignant tumor cells compared to their relatively low physiological expression. The same applies for cyclooxgenase-2 (COX-2), which is constitutively expressed in the kidney, brain, thymus and gastrointestinal tract, but induced in microglia, macrophages and synoviocytes during inflammation. This puts TSPO and COX-2 in the spotlight as important targets for the diagnosis of inflammation. Imaging modalities, such as positron emission tomography and single-photon emission tomography, can be used to localize inflammatory processes and to track their progression over time. They could also enable the monitoring of the efficacy of therapy and predict its outcome. This review focuses on the current development of PET and SPECT tracers, not only for the detection of neuroinflammation, but also for emerging diagnostic measures in infectious and other non-infectious diseases such as rheumatic arthritis, cancer, cardiac inflammation and in lung diseases.

## 1. General Introduction

This review aims to examine translocator protein (TSPO) and cyclooxygenase (COX-2) as biomarkers which are not limited to neuroinflammation but also associated with other types of inflammation attributed to tumor proliferation, autoimmune diseases, pulmonary diseases and myocarditis. More specific targets for some of the afore mentioned inflammatory conditions have already been developed to overcome apparent limitations of TSPO as a biomarker in brain imaging. These limitations include intersubject variability [1], inability to differentiate between the detrimental and beneficial effects of inflammation and inability to differentiate the roles of microglia and astroglia, which both play important roles in neuroinflammation [2]. Other targets have been identified to serve as potential PET biomarkers such as the macrophage colony-stimulating factor 1 receptor (CSF1R) that is mainly expressed in microglia [3] or MAO-B receptors, which are highly expressed in astroglia [4]. The purinergic receptors P2X_7_ (proinflammatory effects) [5] and P2Y_12_ (neuroprotective effects) [6] are also expressed in microglia and perform different roles in the modulation of inflammation. These biomarkers are discussed in detail in other reviews [7,8]. The goal of this review is to highlight the versatility of the TSPO and COX biomarkers, and also the efforts in optimizing not only the physico-chemical properties of the respective radiotracers but also their radiolabeling especially for the more elusive TSPO. Other obstacles to the imaging of TSPO and COX-2 will be addressed in pertinent parts of this review. 

## 2. Translocator Protein (TSPO) 

### 2.1. Introduction

Translocator protein (TSPO) 18 kDa was originally described as a peripheral benzodiazepine receptor (PBR), a secondary binding site for diazepam, in order to differentiate it from the central benzodiazepine receptor (CBR), which constitutes a part of the GABA_A_ receptor complex. The discovery of PBR in the central nervous system (CNS) (located in the ependymal and glial cells) proved that these receptors are not only be found in the periphery [9]. TSPO is a highly hydrophobic protein with five transmembrane helical domains, located in the outer membrane of mitochondria [9,10,11,12]. It is best known for its role in cholesterol transport from the outer mitochondrial membrane to the inner membrane, where cholesterol is metabolized to pregnenolone, a product necessary for steroidogenesis. Consequently, TSPO is found in high concentrations in steroid-producing organs, such as the adrenal cortex, the testes and the ovaries, but also in the heart and kidneys [10,11]. It is important to note that TSPO is expressed at very low levels in resting microglia in the healthy brain [11].

The emerging interest in TSPO as a PET-imaging target is attributed to its pronounced upregulation during inflammation. In neuroinflammation, for instance, pro-inflammatory activation of the microglia upon acute injury or during progressive neurodegeneration has been observed and for a while now has been believed to contribute to dementia as a cause and a downstream effect [8,13,14,15]. The pathological changes seen in neurodegenerative diseases (NDD), such as abnormal accumulation of protein aggregates as found in Alzheimer’s disease (AD), dementia with Lewy bodies (DLB), frontotemporal dementia (FTD) or Parkinson’s disease have been linked to microglial and astroglial activation into proinflammatory states [16,17,18,19]. Immunohistochemical (IHC) staining together with PET-imaging have further confirmed the increased expression of TPSO in a variety of animal models with neuroinflammatory conditions, as found in NDD, and this colocalizes to reactive astroglia and activated microglia [14,15]. 

Despite limitations such as high lipophilicity [20], result-confounding radiometabolites [21] and most importantly, intersubject variability, which is a consequence of the TSPO rs6971 single-nucleotide polymorphism (SNP) [1,14] associating with existing TSPO PET-tracers [8], there are still ongoing efforts to develop novel PET tracers with optimal lipophilicity as well as diminished susceptibility to the rs6971 SNP [14]. Fortunately, the four domains needed for interactions of ligands with TSPO have already been discovered, and they include one hydrogen-bond donor group and three major lipophilic regions [12]. Based on this motif, several small-molecule TSPO radiotracers have been synthesized. Moreover, PET studies have revealed high TSPO binding in various dementias, pointing at the great potential of using TSPO as a biomarker for exploring the contribution of neuroinflammation in the pathogenesis of dementia and monitoring the progression of neurodegenerative diseases [15]. Interestingly, recent evidence suggests that increased neuronal activity is also accompanied by increased TPSO expression, which could provide insight into some disease processes [16]. Consequently, the detection of neuroinflammation via TSPO detection may not be cell type-specific and may also involve non-inflammatory neuronal activity [17].

### 2.2. First Generation of TSPO Radiotracer

#### [^11^C](R)PK11195

The first nonbenzodiazepine-type ligand discovered and the most studied PET TSPO tracer worldwide (Figure 1) is a carbon-11-labeled isoquinoline carboxamide derivative identified in the early 1980s as a TSPO antagonist [18,19]. In vitro binding assays using [^3^H]PK11195 showed a K_d_ of 1.4 nM in rat brain tissue and a K_d_ ranging from 4.3 to 6.6 nM in human brain tissue [20]. 

Despite its high lipophilicity (log P 3.4) [21], it still remains the most widely used TSPO radiotracer due to its apparent lack of susceptibility to the rs6971 SNP [22,23]. Another important shortcoming of the tracer is that it can only be radiolabeled with the short-lived carbon-11 (20 min half-life), which is only available in centers with an on-site cyclotron and fundamentally limits its usage in routine clinical practice. High plasma protein binding and non-specific binding, both consequences of its high lipophilicity, result in both poor brain uptake and poor signal-to-noise ratio as observed in PET studies in healthy controls [11,13,24]. Moreover, its low BP_ND_ (non-displaceable binding potential) and low receptor affinity were also attributed to its high lipophilicity, which may also facilitate unspecific binding of the tracer in the periphery [24]. This was confirmed via comparative blocking experiments carried out in the same healthy controls. These studies revealed that [^11^C](R)PK11195 has a low BP_ND_ of 0.8, which roughly corresponds to a low signal-to-noise ratio [15], which means that it does not have the sensitivity needed to detect mild neuroinflammation, and therefore [^11^C](R)PK11195 cannot be used for early diagnosis and the tracking of subtle changes in disease therapy.

So far, clinical studies carried out with [^11^C](R)PK11195 have provided mixed results. Some initial studies reported high brain TSPO PET signal compared to controls in patients with AD, Parkinson’s disease (PD) and amyotrophic lateral sclerosis (ALS) and those at risk of Huntington’s disease (HD) [25,26,27,28,29,30,31,32,33]. In early PD patients, the tracer detected activated microglia in the nigrostriatal brain regions. Additionally, it showed a difference in the distribution of microglia in PD and Lewy body dementia (LBD), which supported the conclusion that these diseases proceed differently based on nigrostriatal and cortical distribution of the activated microglia [34,35]. 

### 2.3. Second-Generation TSPO Tracers

The high lipophilicity of [^11^C](R)PK11195 and its related limitations already mentioned led to the development of a second generation of TSPO tracers, which include but are not limited to the following: the phenoxyarylacetamide derivatives [^11^C]PBR28 and [^18^F]FEPPA; and pyrazolopyrimidines [^11^C]DPA-713, [^123^I]/[^124^I]/[^125^I]DPA-713, [^18^F]DPA-714 (PBR099) and [^18^F]F-DPA (Figure 1) [36]; most of these have been studied in humans, especially in mild cognitive impairment (MCI) and AD patients and in patients with Parkinson’s disease [6,10,11].

Reduced lipophilicity accompanied by improved signal-to-noise ratio was observed with this generation of TSPO tracers, which facilitated the discovery of the human rs6971 SNP. The rs6971 SNP results in a non-conservative substitution at the 147th amino acid of TSPO, where alanine is exchanged for threonine (A147T). Codominant expression of this genetic trait results in three genotypes, low- (LABs), mixed- (MABs) and high-affinity binders (HABs), as was discovered with [^11^C]PBR28 and [^11^C]DPA-713. HAB subjects are homozygous (A/A) for wild-type (WT) TSPO, while MABs and LABs are heterozygous (A/T) and homozygous (T/T), respectively, for the A147T TSPO [1,14]. The tracers bound with low affinity in approximately 5–25% of human donors (LABs), around 50–65% of HABs and about 30% of MABs. In the MAB group, some ligands showed a two-site binding affinity, while others displayed a K_i_ between that of the HAB and LABs [14,37,38,39]. 

For [^11^C]PBR28, in which this feature was first discovered, there was a 40-fold difference between the HABs and LABs. The polymorphism had such a huge effect on the binding of [^11^C]PBR28 that subjects who were homozygous LABs showed no detectable brain signals from the tracer [15]. The SNP might not have been detected with [^11^C](R)PK11195 due to its low specific binding in the brain; however, sensitivity to the rs6971 SNP was observed in peripheral organs with much higher expressions of TSPO, such as the heart and the lungs [22,23]. 

The discovery of the rs6971 SNP in the TSPO gene in the second-generation TSPO led to a change in the design of clinical experimentation, with studies performed based on genotype tests, or with the exclusion of LABs, which comprise ~10% of the population [13,14,15,40]. Genotyping, whereby the SNP (rs6971) within the TSPO gene on the chromosome 22q13.2 is genotyped, seems to be a useful tool for the accurate separation of the three TSPO genotypes [41,42].

These new paradigms led to more consistent findings in studies of neuroinflammation in AD patients, based on increased tracer retention than with the first-generation tracer [13]. Moreover, in comparison to controls, there were consistently higher TSPO PET brain signals in patients with AD, ALS, MCI, PD and multiple sclerosis (MS) [13,14,41]. Full-blockade (receptor/transporter) studies conducted in monkeys in another study showed that the evaluated second-generation TSPO radiotracers had a higher specific binding than [^11^C](R)PK11195. For example, [^11^C]PBR28 showed a specific binding 80 times higher than that of [^11^C](R)PK11195 in vivo in monkey brains [22]. Further studies with [^11^C]PBR28 and other TSPO tracers in healthy subjects based on baseline and blocking studies revealed the BP_ND_ of the TSPO tracers in HABs. [^11^C]DPA-713 showed the best BP_ND,_ 7.3, followed by the third-generation TSPO tracer, [^11^C]ER176, with a BP_ND_ of 4.2, and [^11^C](R)PK11195, with a BP_ND_ of 0.75, which was 1.5 times lower than that of [^11^C]PBR28, with a BP_ND_ of 1.2 [15,23,40]. 

Compared to [^11^C](R)PK11195, [^11^C]DPA-713 not only provided the highest signal-to-background ratio in HAB, but also showed an accumulation of radiometabolites in the brain, which was consistent with increased distribution volume (V_T_). Although brain radiometabolites are usually detrimental to the quantification of PET images, in this case, their effect was diluted due to the high specific binding of [^11^C]DPA-713 in HABs and MABs and further reduced by the relatively low concentration of brain radiometabolites (compared to the general brain uptake). The effects of the brain radiometabolites, however, cannot be ignored in the case of LABs, for whom this could pose significant impediments [23], since there are barely any specific signals in the first place.

In a PET study using a herpes simplex encephalitis virus-1 (HSV-1) rat model, a fluoroethylated analog of [^11^C]DPA-713, namely [^18^F]DPA-714, showed lower non-specific binding than [^11^C](R)PK11195 and slightly lower specific binding than its predecessor [^11^C]DPA-713 in infected rat brains. To determine the specific binding, the average tissue/plasma ratio of the rat model pretreated with non-radioactive PK11195 was subtracted from the tissue/plasma ratio of the control and rat models separately for each rat. Nevertheless, the brain uptake of [^18^F]DPA-714 was lower than that of [^11^C]DPA-713 and [^11^C](R)PK11195 in both healthy and infected brain tissues, but with specific uptake comparable to that of [^11^C](R)PK11195. Despite its relatively low performance, [^18^F]DPA-714 was found to be an agonist at the TSPO receptor, which might take advantage of the high-affinity state of TSPO resulting in improved binding of the tracer to TSPO in chronically activated microglia cells during progressive neurodegeneration [42].

[^18^F]F-DPA, like [^18^F]DPA-714, is another interesting fluorine-18-labeled pyrazolopyrimidine tracer, but unlike the latter, the fluorine atom is bound to a Csp2 aromatic carbon. The tracer was discovered as early as 2001 by Selleri et al. [43]. Its affinity to TSPO (1.7 nM) and selectivity over CBR (>1 μM) in a competitive binding experiment were comparable to those of its fluoroethylated analog [^18^F]DPA-714, which were 0.91 nM and >1 μM respectively [44].

Until recently, the radiosynthesis of [^18^F]F-DPA has been fraught with difficulties, which resulted in low radiochemical yield (RCY) and low molar activity (A_m_). Radiosynthesis via [^18^F]fluoride resulted in <3% RCY, probably owing to its lack of ring activator. Keller et al. [44] were able to improve RCY via electrophilic radiofluorination via carrier-added [^18^F]F_2_, but improved RCY (15%) was accompanied by a low A_m_ (7.8 GBq/μmol). Wang et al. [45] were able to obtain the tracer at a higher RCY (45%) with [^18^F]fluoride by using an optimized spirocyclic iodonium ylide (SCIDY) precursor. The A_m_ obtained by Wang et al. [46] was also higher (96 GBq/μmol) compared that obtained by Keller et al. [44]. Automation of synthesis [46] further improved RCY to 15.6% (non-decay corrected), with a >5-fold increased A_m_. Further optimization of synthesis conditions using the same precursor resulted in up to almost 20% RCY [41]. Although [^18^F]DPA-714 obtained with low A_m_ still demonstrated the ability to distinguish between healthy and AD transgenic mice [47], it might not be able to detect subtle changes in certain cases. 

[^18^F]F-DPA displayed saturable specific binding to TSPO in rodent models of neuroinflammation (AD and cerebral ischemia). Around 1.2–1.6 times higher brain uptake of the tracer was seen in AD mice compared to age-matched controls. It also demonstrated high uptake into the ischemic brain hemisphere in rats with focal cerebral ischemia. Uptake was diminished (by 80%) when co-administered with PK11195 (3 mg/kg), which indicated specific in vivo uptake and low non-specific binding. Specific binding in autoradiography in vitro was also determined via co-incubation with PK11195. In a simplified reference tissue model, BP_ND_ was determined to be 2.5 times higher than that of [^11^C]PK11195 [45]. Compared to [^18^F]DPA-714, [^18^F]F-DPA displayed higher in vivo stability in the plasma (2.5-fold) at 90 min post-injection (p.i). At the same time point, the brain content of the unchanged [^18^F]F-DPA was 1.7 times higher compared to [^18^F]DPA-714 in healthy rats [44]. This was expected, since the aryl-^18^F bond present in [^18^F]F-DPA is stronger than the Csp3-^18^F bond in [^18^F]DPA-714.

López-Picón et al. [48] demonstrated in comparative PET studies that [^18^F]F-DPA might be a better tracer than [^18^F]DPA-714 and [^11^C]PBR28 in detecting low levels of inflammation. Compared to the other two tracers, it displayed higher brain uptake and washout as well as higher transgenic-to-wild-type standardized uptake value ratios (SUVR) than the SUVR obtained from the cerebellum.

### 2.4. Third-Generation TSPO Tracers

Even though genotypic stratification has proved helpful in clinical studies with second-generation tracers, there is still the need to develop PET TSPO tracers which do not discriminate between the genotypes, i.e., between wild-type and A147T mutations. 

With this in mind, Zanotti-Fregonara et al. [12] developed [^11^C]ER176, a quinazoline analog of [^11^C](R)PK11195, (Figure 2) with very diminished sensitivity to the human SNP rs6971 [12,49], and a BP_ND_ ratio of 1.3:1 in HABs and LABs. This affinity ratio was even higher for [^11^C]PBR28 at 55:1 [49]. This means that LABs do not need to be further excluded in clinical studies with this tracer; however, due to a slight sensitivity to the genotype, the V_T_ values have to be corrected for genotype a posteriori [40]. Another unique advantage of [^11^C]ER176 over the other TSPO PET tracers, apart from having a BP_ND_ of 4.2—the second highest, after [^11^C]DPA-713—was the absence of brain radiometabolites, which resulted in more time-stable values of V_T_ in both HABs and LABs after the blockade of TSPO receptors by XBD173 [40]. So far, clinical studies have been carried out only in healthy subjects; hence, there is still a need to perform these studies in patients with neuroinflammation [14]. 

In addition to the already-mentioned properties, [^11^C]ER176 showed a higher affinity (3.1 times) in rat kidney mitochondrial membranes. Moreover, it exhibited less lipophilicity (1.3 times) in comparison to its direct analog [^11^C](R)PK11195 [12], which resulted in higher unbound fractions in the plasma, improved brain uptake and PET signal (in monkey brains) and higher specific binding confirmed by up to 80% signal blockade after the administration of cold (R)PK11195 [40]. There is an ongoing clinical study, which aims to assess the usefulness of [^11^C]ER176 for the accurate quantification of microglial activation in patients with AD [50]. 

To make up for the short half-life of the carbon-11-labeled [^11^C]ER176, its fluorine-18-labeled analog, [^18^F]BIBD-239, was developed by Chen et al. [51]. [^18^F]BIBD-239 (IC_50_ 5.24 nM) showed a slightly improved affinity for TSPO compared to [^11^C]ER176 (IC_50_ 5.94 nM). [^18^F]BIBD-239 also showed binding modes and sites similar to those showed by Ala147-TSPO and Thr147-TSPO in theoretical simulation, a characteristic which suggests low sensitivity to the rs6971 polymorphism. It showed nearly 2 %ID/g in an ICR mouse model and was also able to clearly detect focal cerebral ischemia in Sprague–Dawley (SD) rats with mild focal ischemia in PET studies in vivo. In vitro autoradiography experiments with the ischemic rat brain correlated with the PET studies. Clear tumor PET images with [^18^F]BIBD-239 were also obtained in a GL261 mouse model. Additionally, at 60 min p.i., no metabolites of [^18^F]BIBD-239 were found in the brain, even though the biotransformation of the tracer was rapid in the periphery, as indicated via low plasma content of the tracer at 60 min p.i.

The [^18^F]fluorine-labeled TSPO tracer flutriciclamide [^18^F]GE-180, a tricyclic indole derivative discovered by Wadsworth et al. [52], also belongs to this tracer generation (Figure 2). The S-enantiomer displayed superior properties in comparison to the D-enantiomers in terms of binding affinity and pharmacokinetics, with 4.4-fold greater affinity to the target and a faster clearance from the striatum (a low expressing region) and a relatively high percentage of the parent tracer (94%) in the brain at 60 min p.i. [52]. Flutriciclamide also showed better imaging characteristics compared to [^11^C](R)PK11195 and [^18^F]fluorine-labeled DPA-713, namely [^18^F]DPA-714, in preclinical studies [53] but failed to show the extent of microglia/macrophage activation as accurately as [^11^C]DPA-713, which also afforded earlier detection of inflammation [54]. 

In healthy subjects, however, [^18^F]GE-180 showed poor imaging properties, with a very low brain uptake and an almost flat time activity curve, and due to its high uptake in blood vessels, kinetic modeling was difficult [55,56,57]. In comparison to [^11^C]PBR28, it displayed a 20-fold lower V_T_, as a result of its inability to penetrate the brain from blood vessels. This, however, inadequacy belies its log D of 2.95, which suggests good brain penetration. This led to the conclusion that the tracer is a substrate of efflux proteins at the BBB [56]. Zanotti-Fregonara et al. [56] pointed out that this was at odds with the high lesion-to-background ratio observed in brain tumors and MS lesion sites, which suggests that the disruption of the BBB, which is a hallmark of these pathologies, likely facilitated the brain entry of both the tracer and its radiometabolites. This was supported by gadolinium uptake, especially into multiple sclerosis (MS) brain tissue. This means that the signals observed were nonspecific [58]. Albert et al. [59] countered that the high uptake of the tracer in MS lesions, in glioma and in other neurological diseases correlates with disease severity and outweighs its underperformance in healthy subjects, for whom the tracer was not developed [59]. A prior study carried out by Sridharan et al. [60] showed that the TSPO ligand XBD173 competitively displaced [^18^F]GE-180 in brain tissue of MS patients; hence, the uptake into MS lesions might be specific after all [59,60].

Another important criticism of the tracer referred to the proposed in vivo insensitivity to TSPO polymorphism, which, surprisingly, was owed to a 15:1 affinity ratio in vitro between HABs and LABs, which was less than what was displayed by [^11^C]DPA-713. It was argued by Zanotti-Fregonara et al. [58] that the poor image quality of the PET, a consequence of low brain uptake, made such a glaring difference undetectable in vivo [37]. In response, Albert et al. [59] contended that lack of allelic discrimination in vivo should be a sought-after quality for a TSPO tracer but not the reverse. However, further experiments need to be carried out in order to generally understand allelic dependence on affinity of TSPO tracers [60]. Also, further experiments will be carried out on brain tumor patients in order to spatially compare [^18^F]GE-180-PET with histopathological analyses of tissue samples from stereotactic biopsies [59].

In a study by López-Picón et al. [61] the longitudinal relationship between the deposition of Aβ and neuroinflammation in the APP23 AD mouse model with both the TSPO-tracer and the Aβ tracer [^11^C]PIB was determined. It was observed that there was a clear age-dependent increase of [^11^C]PIB with increased β-amyloid aggregation in the frontal cortex (FC), parietotemporal cortex (PTC), thalamus and hippocampus [44]. Like [^11^C]PIB, [^18^F]GE-180 convincingly correlated neuroinflammation and Aβ deposition in the PTC and thalamus, but [^18^F]GE-180 displayed minimal binding in areas of early amyloid deposition (FC and hippocampus). Moreover, the binding of [^18^F]GE-180 reached a plateau earlier in the pathogenesis of the AD in contrast to [^11^C]PIB. This is an indication that [^18^F]GE-180 might be a useful tracer for early detection of pathological neuroinflammation in AD, but not useful for the long-term tracking of disease progression [61]. 

A more recent longitudinal PET study (supported by autoradiography) carried out by Holzgreve et al. [62] with [^18^F]GE-180 in an orthotopic syngeneic GL261 glioblastoma (GBM) mouse model demonstrated a continuous uptake of the tracer over time, which overlapped with contrast-enhancement in CT and tissue-established observations. Therefore, it has been concluded that the tracer might prove useful for the imaging of brain tumors.

An analog of [^18^F]GE-180 was recently developed by Qiao et al. [9] by exchanging one of the ethyl substituents on the amide nitrogen for a butyl substituent. Varying the substituents on this functional group is believed to influence the HAB/LAB ratio. Like its predecessor [^18^F]GE-180, the S-enantiomer based on a rat heart TSPO assay showed a higher binding affinity (1.04 nM) than the R-enantiomer (21-fold less). 

[^18^F](S)GE-387 (Figure 2) showed little loss in affinity to A147T TSPO when evaluated for sensitivity to TSPO polymorphism in an assay using human embryonic kidney cell lines, with a LAB/HAB ratio of 1.3, which is comparable to that of [^11^C](R)PK11195 (1.2). A further in vivo PET analysis in healthy rats showed a modest distribution of the racemic mixture of both enantiomers of the tracer in the brain; hence, there is a possibility that in pathology there might be a higher uptake of the tracer. Further, biological evaluation is still being carried on both enantiomers of the tracer [9]. 

The efficacy of [^18^F](S)GE-387 as a useful TPSO tracer was additionally evaluated by Ramakrishnan et al. [63] in a lipopolysaccharide (LPS)-induced (10 µg/4 µL) neuroinflammation rat model corroborated the claims by Qiao et al. [9]. They observed that the tracer can discern inflamed from healthy brain regions three days after the injection of LPS in the brain. Likewise, they confirmed low sensitivity of the tracer to the TPSO rs6971 polymorphism in genotyped human brain tissue.

Other recently reported third-generation TPSO tracers include [^18^F]CB251 and [^18^F]BS224. The former is a structurally modified version of the second generation tracer [^18^F]PBR111. [^18^F]PBR111 exhibited specific binding to TSPO, which facilitates both in vivo visualization and quantification of neuroinflammation [64,65]. Unfortunately, it is sensitive to the TPSO polymorphism [65]. [^18^F]CB251, on the other hand, in addition to showing high cellular uptake in modified cells and LPS-induced neuroinflammation mice, showed comparably low sensitivity to the rs6971 polymorphism and, based on PET/MRI studies, high sensitivity to changes in the severity of neuroinflammation [66].

[^18^F]BS224 is a direct analog of [^18^F]CB251 (Figure 2), with the [^18^F]fluoroethyl group exchanged for a Csp2[^18^F]fluorine atom, that is, a fluorine atom directly bound to the benzene ring. This was carried out in a bid to reduce in vivo defluorination observed in the fluoroethylated [^18^F]CB251. In vitro competitive inhibition assays with membrane proteins showed that [^18^F]BS224 has low sensitivity to the rs6971 polymorphism, with an almost equal affinity to HABs (IC_50_ 0.59 nM) and LABs (IC_50_ 0.45 nM): LAB/HAB ratio 0.76. Moreover, the LAB/HAB ratio of [^18^F]BS224 was comparable to that of [^11^C]PK11195 (0.83), which is believed to be insensitive to the polymorphism. Although the fluoroethylated [^18^F]CB251 showed an overall higher affinity to both LABs and HABs, the LAB/HAB ratio was nonetheless higher (1.14), making it, in regard to the other two, the worst in this respect. Moreover, [^18^F]BS224 in rats models of LPS-induced inflammatory and ischemic stroke clearly labeled inflammatory lesions with high BP_ND_ of 1.43 ± 0.17 and 1.57 ± 0.37, respectively [67]. 

There has been marked improvement in the susceptibility of the TSPO tracers to the SNP from the second to the more recent generation of tracers; however, this still remains a factor that has to be considered in the further development of TSPO tracers. Therefore, efforts are still being made to better understand the difference in the binding requirements for both the wild type and the A147T variant [68,69,70] by carrying out high-throughput screening of ligands [71], identifying other binding sites in the TSPO protein [72], and increasing understanding of in vivo kinetics, physiological modulators of binding and brain penetration of radiometabolites, especially the effects of these radiometabolites on PET signals in LABs [40,73]. A list of the reported TSPO radiotracers is presented in Table 1.

## 3. The Cyclooxygenase-2 (COX-2) Enzyme

### 3.1. Introduction

Cyclooxygenase-2 is one of the two COX isozymes also known as prostaglandin-endoperoxide synthase (PTGS) [74], which is induced in response to inflammation and pain in the body [75,76,77]. The other isoform of the enzyme, COX-1, is predominantly expressed constitutively and plays a role in the maintenance of organ homeostasis, including gastric cytoprotection, and maintenance of renal function [78,79,80,81]. Recent findings suggest that COX-2 is also constitutively expressed in the kidneys, where it plays a role in modulation of vascular tone in addition to regulation of hydric balance [81]. While constitutive COX-2 mRNA expression can be detected in the gut and kidneys, high COX-2 mRNA levels are found in the normal rat brain [82] in which it was also discovered. Moreover, strength and endurance training increases brain COX-2 expression, which is involved in neuronal plasticity and learning [83].

Recently, however, Shrestha et al. [80] observed a lack of displaceable binding of [^11^C]MC1 (IC_50_ COX-2 3 nM; selectivity over COX-1 > 3000-fold [84]) to COX-2 in monkey brains at baseline, whereas specific binding was observed in the ovary [80], which corroborated previous results obtained by Kim et al. [85]. These findings suggested that COX-2 might not be expressed in a healthy human brain at all, which makes it an interesting target for the evaluation of inflammatory processes in the brain.

COX-2 is present intracellularly on the luminal side of the smooth endoplasmic reticulum, the Golgi apparatus and the nuclear membrane [86,87,88], with its active site located on the membrane-bound portion of the enzyme. This is a constraint in terms of drug delivery, since only highly lipophilic ligands can transverse both the cell and the organelle membranes. Even so, very lipophilic ligands are known to bind non-specifically and suffer from slower clearance; hence, longer waiting times are needed for the ligands (radiotracers) to be cleared from non-target tissues in order to obtain better target-to-background ratios [89,90].

Moreover, many premalignant neoplasms are marked by the overexpression of COX-2 [87], with elevated upregulation in a variety of cancers such as breast, colorectal and gastric cancers [78,87,89]. It also plays a role as a biomarker and effector enzyme in neural damage both after brain trauma and in the ageing brain and associated pathological conditions [87], such as in NDDs such as PD [91] and AD. 

The molecular mechanisms underlying COX-2 expression in certain types of cancers and inflammation have been widely investigated. However, there still remains a controversy regarding the exact role of COX-2 in NDDs and carcinogenesis, which is further aggravated by the discrepancies between the anti-cancer effects of some COX-2 inhibitors in vitro and their lack of therapeutic efficacy in vivo. With an effective tracer for in vivo imaging, it will be possible to understand the elaborate role of COX-2 in the pathogenesis of various diseases and to confirm or disprove hypotheses regarding its contribution and regulation in different pathologies [78,87,89]. 

So far, several COX-2 imaging agents have been synthesized both for PET and SPECT imaging, with only [^11^C]MC1 so far making it to clinical trials [76]. The fact that the evaluation of COX-2 expression is only possible via elaborate ex vivo analyses makes the development of tracers for this enzyme quite challenging; for instance, COX-2 mRNA and protein are unstable in vitro and undergo quick degradation [78]. 

In this review, some interesting carbon-11- and fluorine-18-labeled COX-2 PET tracers together with some iodine-125 and 123 and Tc-99m-labeled SPECT tracers are discussed. 

### 3.2. PET COX-2 Radiotracers

#### 3.2.1. [^11^C]Carbon Labeled COX-2 Tracers

Earlier in the development of [^11^C]carbon-labeled COX-2 tracers, mostly FDA-approved COX inhibitors were used, even though among these, celecoxib is the only COX-2 inhibitor still approved for use in clinical practice. Rofecoxib was withdrawn from the market due to increased cardiovascular risks after long-term use of high doses [92]. The latter, of course, does not constitute a problem, since radiotracers are used in the short term and at sub-therapeutic concentrations with insignificant pharmacological effects [78]. These COX inhibitors were radiolabeled to enable a smooth transition into clinical practice and to facilitate the translation of data obtained via PET studies into data of clinical relevance. 

Celecoxib (Figure 3) is a COX-2 inhibitor currently still applied in clinical practice [93]), and its derivatives were deemed lead structures for the development of COX-2 radiotracers. In an experiment to evaluate drug transport mechanisms in biliary excretion, carbon-11-labeled celecoxib (K_i_ COX-2 40 nM; selectivity over COX-1 425-fold) [94] was found to have low in vivo stability. It was quickly metabolized into carboxylic acid and hydroxymethyl-containing products [95], and due to the presence of the sulfonamide functional group present in the molecule, there was high blood retention of the tracer. It is believed that although the binding site of carbonic anhydrase (CA) and COX-2 have similar shapes, the presence of zinc in the catalytic site of the former increases its affinity to sulfonamide derivatives, perhaps due to the strong bond between the zinc-containing catalytic site and the primary amine in the sulfonamide moiety [95]. Other derivatives of celecoxib were disqualified for in vivo imaging of COX-2 due to high unspecific binding as seen, for example, in tracers developed by Fujisaki et al. [96] and suboptimal affinity to the target as seen in the tracers developed by Gao et al. [97], probably due to the compromise of the sulfonamide group, a functional group necessary for affinity to the target.

[^11^C]carbon-labeled rofecoxib (Figure 3) with K_i_ COX-2 18–44.6 nM and K_i_ COX-1 50,000 nM performed slightly better than the celecoxib tracers, but not in all experiments: rofecoxib showed specific binding in healthy rat brains, but unfortunately the same result could not be replicated in inflammation models, a failure partly blamed on the inadequacy of the models, especially the sterile inflammation model [74,98].

Unlike the other aforementioned COX-2 tracers, [^11^C]MC1 (Figure 4) contains a 6-membered heterocyclic pyrimidine ring in the center of the molecule instead of a 5-membered ring and a methanesulfonyl moiety instead of the CA-susceptible sulfonamide group. It showed good brain uptake in monkey brains, with radioactivity peaking in the brain at 2.9 SUV at 2 min post-injection (p.i.), which quickly reduced to 1.16 SUV at 40 min p.i. [84]. Subsequent examination of the tracer in LPS-treated rhesus macaques (LPS injected into their brains), the results from two patients with rheumatoid arthritis as well as two healthy participants revealed that [^11^C]MC1 can be used to evaluate inflammation both in the brain and in the periphery [80]. This makes it the first PET radioligand that was able to successfully image and quantify COX-2 expression/regulation in vivo. Recruitment for clinical trials with this ligand for PET imaging of cyclooxygenases in dementia, rheumatoid arthritis and myositis are ongoing [99].

**Figure 3 ijms-24-17419-f003:**
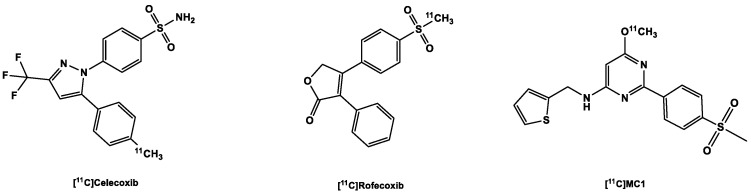
Structure of ^11^C-labeled COX-2 tracers.

#### 3.2.2. [^18^F]Fluorine-Labeled COX-2 Tracers

As with the [^11^C]carbon-labeled COX-2 tracers, the aim was to label COX-2 inhibitors already used in clinical practice, without structural alteration. Hence, celecoxib was radiofluorinated to obtain an [^18^F]trifluoromethyl analog [^18^F]1 (Figure 4) via nucleophilic substitution reaction (S_N_Ar) using a bromodifluoromethyl precursor developed by Prabhakaran et al. [94]. PET scans in rats and baboons showed a high in vivo instability of the tracer partly due to high defluorination confirmed by high bone uptake compared to the heart and brain in rats. Although defluorination was slower in baboons, there remained only 17% of the parent tracer in the plasma at 60 min p.i. Although the rate of biotransformation of the tracer might be reduced in humans owing to their slower metabolic rate compared to both mammals [100], this nevertheless impedes further perspectives on the development of this tracer for in vivo imaging.

To counter in vivo defluorination, Uddin et al. [101] developed another celecoxib analog, [^18^F]2, for which a K_i_ of 160 nM to COX-2 and a K_i_ of 80 nM to COX-1 were reported in free enzyme assays and a K_i_ of 80 nM to COX-2 in whole cell assays. In contrast, this analog bore a [^18^F]fluoromethyl group instead of a [^18^F]trifluoromethyl group [101]. The specificity of the tracer was confirmed in the following experiments: A PET study comparing the uptake of the tracer in both inflamed (carrageenan-treated rat paws) and non-inflamed tissues (non-treated). It showed a 1.53-fold increase in the former over the non-treated paws [101];Pre-dosing with celecoxib (10 mg/kg), which significantly decreased tracer uptake in inflamed rat paws (there was only a 1.7-fold decrease in uptake [101];Experiments in COX-2 null mice further confirmed the specificity of the tracer; there was no increased tracer uptake in the inflamed carrageenan-treated paws of these mice compared to controls, i.e., non-inflamed carrageenan-treated paws of the same COX-2 knock-out mice with a ratio of 1.08. This contrasts significantly with the uptake in the inflamed paws of wild-type mice versus the control paws (1.48) [101];Results obtained using nude mice with both COX-2-positive 1483 HNSCC tumors and COX-2-negative HCT116 tumors suggest that the difference in the uptake in both tumor types correlates with the difference in their expression of COX-2 (3 times higher in the COX-2-positive tumor). The blocking of the COX-2 active site in the former prevented the binding of the tracer; compared to the control, tumor-to-muscle ratio was nearly three times lower [101].

Importantly, it was observed that [^18^F]2 underwent minimal defluorination in vivo [101]. Compared to the carbon in the fluoromethyl moiety in [^18^F]2, the geminal carbon in [^18^F]1 bears a higher positive partial charge despite having a carbon-fluorine bond stronger than that in [^18^F]2 [102]. The electron deficiency of position C3 in the pyrazole moiety to which the fluoro-substituted methyl moieties are bonded [103] further increases the positive partial charge on the geminal carbon and vice versa. This makes both the geminal carbon and the carbon in position C3 more susceptible to nucleophilic attacks [104] and eventual defluorination in vivo. 

The valdecoxib analog [^18^F]3 (Figure 4) also underwent rapid defluorination in mice in vivo, even though it also bears its fluorine atom as in [^18^F]2 (Figure 4), that is, in a [^18^F]fluoromethyl moiety (Figure 4), which underwent minimal in vivo defluorination in mice [101]. It, however, contains an isoxazole ring whereas [^18^F]2 contains a pyrazole ring. This difference might have left [^18^F]3 with a weak metabolic spot [105]. The weak N-O bond in the isoxazole ring might make the tracer more susceptible to in vivo metabolism [106]. In this study, however, the blood pool retention of [^18^F]2 was not reported, although it also bears a sulfonamide moiety as well which is known to have a sub-nanomolar affinity to CA [107]. Further studies with the tracer are still needed to determine its efficacy as an in vivo imaging agent for detection of COX-2.

Based on the lead structure ([^18^F]pyricoxib) (Figure 4) which is in turn based on a diarylpyrimidine backbone (reminiscent of the aforementioned [^11^C]MC1) originally reported by Swarbrick et al. [108], Tietz et al. [109] recently developed a new class of potent and selective radiofluorinated COX-2 inhibitors. 

They discovered that altering the benzyl group substituent (see [^18^F]pyricoxib Figure 4) results (in position 4) resulted in varying degrees of affinity to COX-2: sterically bulky substituents (phenyl and tert-butyl) in this position gave quite low affinity (>10,000 nM) to the enzyme compared to celecoxib (40 nM). Although still modestly sterically bulky, but strongly electron-withdrawing, the nitro-substituent at this position fared better (86 nM). The less bulky electronegative halo-substituents F (7 nM) and Cl (6 nM) outperformed the nitro-substituent [90,109]. The more sterically bulky Br (48 nM) performed poorly compared to the other two (F and Cl) but outperformed the nitro, phenyl and tert-butyl functional groups. The fact that a methyl substituent (5 nM) and a methoxy substituent (7 nM) outperformed or performed as well as the best halo-substituents is an indication that steric bulk plays a major role in the affinity of the ligands to COX-2, although it could also be seen that the successful halo-substituents are also capable of donating a lone pair of electrons in their resonance forms. Notwithstanding, all the ligands showed more selectivity over COX-1 (>100 µM) than the sulfonamide- and pyrimidine-bearing celecoxib (15 µM). Moreover, it was found that affinity to COX-2 was compromised when the methylsulfone group was exchanged for the sulfonamide group. The corresponding sulfonamide counterparts of select methylsulfone compounds were less potent than the latter; for instance, the 4-fluoro/sulfonamide of [^18^F]pyricoxib showed 5.6 times less potency [109]. 

Further studies were conducted by the same group [79] on the fluorine-18 labeled lead compound ([^18^F]pyricoxib). The tracer showed a high uptake and retention in human colon adenocarcinoma HCA-7 and human colorectal carcinoma HCT-116. There was, however, also sufficient uptake and retention of the tracer in COX-2-negative HCT-116 cells, indicative of a good passive diffusion owing to its high lipophilicity (log P 3.37). In any case, the uptake in COX-2-negative cells was significantly lower than in COX-2-positive HCA-7 cells. Results from IHC analysis confirmed a high expression of COX-2 in both HCA-7 and HCT-116 tumors, with higher detection signals in the former. A CD68 (a marker for tumor-associated macrophages) staining showed that the COX-2-positive staining was not due to infiltration of COX-2-expressing macrophages in inflammatory responses to cancer cell inoculation and tumor growth. 

Uptake of [^18^F]pyricoxib was diminished in a concentration-dependent manner in response to pretreatment of the HCA-7 cells with some COX-2 inhibitors, which serves as proof that cell retention of the tracer was specific. There was, however, still some non-specific intracellular binding, reasons for which were not further analyzed. 

In vivo PET imaging studies in HCA-7 and HCT116 tumor-bearing NIH-III nude mice corroborated the COX-2-mediated tracer uptake and retention in HCA-7 tumors. Pre-administration of 2 mg of celecoxib to each of the mice used in the experiment resulted in a 16% decrease in tumor uptake of the tracer at 1 h p.i. Selective COX-2-mediated uptake of the tracer in HCA-7 tumors was showed in a biodistribution study carried out on celecoxib-treated and control HCA-7 tumor-bearing NIH-III mice, which showed a 50% blocking effect by celecoxib. Nonetheless, there was high uptake in the muscle, which the authors believed was caused by interaction of the tracer with a non-COX target. However, the tracer showed sufficient in vivo metabolic stability in mice, with up to 60% of the parent tracer still present at 120 min p.i. 

More recently, Lebedev et al. [87] labeled a metabolically stable celecoxib derivative 4 (Figure 4) (IC_50_ COX-2 1.7 nM) via an automated electrochemical radiosynthesis route, with the fluorine-18 attached directly to the pyrazole ring, with the intention of improving metabolic stability and specificity [87]. The tracer showed a direct correlation with increasing LPS concentrations in whole cell assays, with a corresponding decrease in uptake after blockage of the active sites by celecoxib (32 µg/mL). An ex vivo biodistribution study in healthy wild-type C57BL6 mice demonstrated the uptake of the tracer in the brain (2.2 ± 0.7 %ID/g) at 60 min p.i., with the bone uptake below that of the blood and the brain and almost the same as the muscle uptake; this proofed a low in vivo defluorination. Dynamic PET/CT scans conducted in healthy C57BL6 mice mice showed good pharmacokinetics, with background reducing within an hour p.i. without any apparent defluorination, as was confirmed by lack of skull/vertebrae retention of the tracer. The results also demonstrated a lack of retention of the tracer in the blood, even though the tracer contains the sulfonamide group, which appeared to be the undoing of previously developed COX-2 tracers. 

The major drawback presently is the low decay-corrected yield of 2% (isolated RCY) and specific activity of 3 Ci/mmol. The authors are still working on improving the above in order to allow for translation to clinical practice [87].

McCarthy et al. [110] also developed an interesting tracer [^18^F]SC58125 (Figure 4), which could be a potential tracer for the imaging of COX-2 activity. It was radiosynthesized to an RCY of 10–20% via the nucleophilic displacement of a trimethylammonium triflate salt using a dedicated microwave cavity. Preliminary in vitro binding studies showed a high higher selectivity for COX-2 with an IC_50_ value of 86 nmol/L compared to that of COX-1 (IC_50_ 100 mol/L). This was confirmed in a cell uptake assay with J774 macrophages stimulated by LPS and IFN-γ for 18 h, in which the uptake of the tracer was increased compared to the control.

Moreover, in an in vivo biodistribution experiment in healthy female SD rats, brain uptake was moderate 0.35 %ID/g at 30 min p.i. At 60 min p.i. there was up to 28.6% decrease; the brain concentration did not change significantly even at 180 min p.i., suggesting some off-target binding in the brain or high binding affinity to constitutively expressed COX-2 [81,111,112]. This was not the case for the iodine-125-labeled [^125^I]IMTP (Figure 5). Even though [^125^I]IMTP is more lipophilic and bulkier (ClogP 5.1, MW 490.0 Da) than [^18^F]SC58125 (ClogP 4.1, MW 383.4 Da), it showed a somewhat more consistent brain clearance from 10 min (0.25 %ID/g) to 180 min (0.10 %ID/g) [113]. However, its bulkiness might have reduced its initial uptake.

At 30 min p.i., the accumulation of the tracer in the kidney was also high, reached 0.6 %ID/g, indicating a high expression of COX-2 in this organ. Bone uptake was also low at this time point (0.22 %ID/g), indicative of low in vivo defluorination, and decreased to 0.14 %ID/g at 180 min p.i., in contrast to the brain pharmacokinetics of the tracer [110].

**Figure 4 ijms-24-17419-f004:**
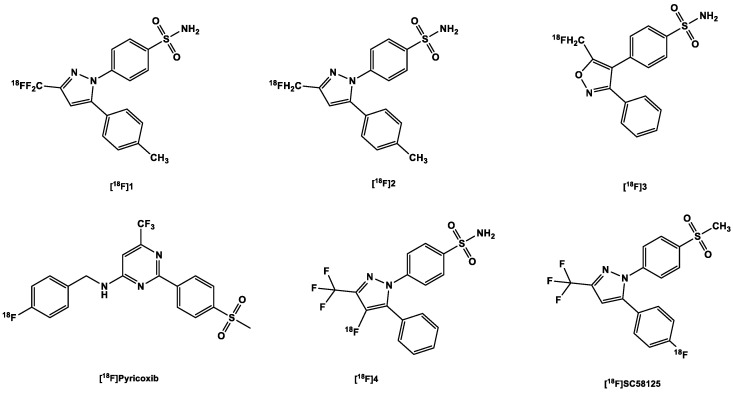
Structure of ^18^F-labeled COX-2 tracers.

### 3.3. SPECT COX-2 Radiotracers

#### 3.3.1. [^123,125^I]Iodine-Labeled COX-2 Tracers

As mentioned earlier, the location of the COX-2 target calls for a tracer with a high lipophilic profile in order to allow the tracer to passively transverse both the cellular and organelle membranes [86,87,88]. Of course, there is the disadvantage that highly lipophilic tracers may bind indiscriminately to tissues in vivo and for this reason require longer waiting times in order to allow for sufficient clearance of the tracers from non-target sites in order for a reasonably high imaging contrast to be achieved [90]. Longer waiting times of up to 4 h require longer-lived isotopes (of which fluorine-18 falls slightly short, much less carbon-11), which would allow for longer imaging experiments, in order to yield better imaging contrast [90]. Moreover, COX-2 inhibitors inhibit COX-2 in a time-dependent manner; hence, the longer-lived the radio-isotope, the better, in order to ensure that binding equilibrium is achieved before the complete decay of the radioisotope [113]. Although some tracers already mentioned in this review may provide acceptable detection results with shorter-lived radio-isotopes, it is still nevertheless preferable to have other options. 

To this end, the gamma emitters ^123^I, ^125^I and ^99m^Tc find useful application. Due to their comparably longer half-lives of 13.2 h, 59.4 days, and 6.0 h, respectively, these radionuclides are easier to handle relative to their positron emitting counterparts. The resolution of SPECT images is nonetheless considerably lower in comparison to PET [78]. In addition, the steric bulk of iodine (a size comparable to that of a phenyl group) [90] makes it one of the less commonly used halogens in small molecules, which is in sharp contrast [80,85] to the relatively small size of fluorine, an isosteric analog of hydrogen [76,81]. Moreover, the steric effects caused by the presence of iodine in small molecules limits its inclusion in binding pocket of the COX-2 active site thereby also reducing affinity [90]. 

Kabalka et al. [114] developed a ^123^I analog of celecoxib (Figure 3): the methyl group was replaced with [^123^I]5 (Figure 5) to retain affinity (IC_50_ COX-2 8.2 nM) and specificity, as halogens are better tolerated in position C4 [114]. Further biological evaluation was carried out by Schuller et al. [115] in nicotine-derived nitrosamine 4-(methylnitrosamino)-1-(3-pyridyl)-1-butanone (NNK)-treated hamsters, which resulted in the induction of adenocarcinomas in their pancreases and lungs. Biodistribution experiments performed on these hamsters showed higher uptake of the tracer in the liver and pancreas and a slightly higher uptake in the lung. Additional whole body imaging with the ^123^I-labeled tracer showed increased uptake in the pancreas in one of the hamsters and the liver of two of the hamsters, which correlated with the IHC staining of the organs. The third hamster showed neither accumulation in the aforementioned organs nor overexpression of COX-2 via IHC. However, the results showed that the distribution of the tracer correlates with the overexpression of COX-2 and that the uptake of the tracer might be mediated by COX-2. These results corroborated the resulted obtained from the in vitro evaluation of the tracer, in which it was observed that there was higher uptake in the NNK-pretreated cells than the non-treated controls. This uptake was diminished after pre-incubation with celecoxib, proof that uptake was due to presence of COX-2 in the cells [78,115].

Kuge et al. [113] in 2006 provided additional information on a compound developed by Kabalka et al. [114] ([^125^I]IATP) (Figure 5) and a methylsulfone derivative ([^125^I]IMTP). Both tracers showed good binding to the COX-2 enzyme ([^125^I]IMTP 5.2 µM and [^125^I]IATP 8.2 µM, respectively) [113] and selectivity over COX-1 (>100 µM), resulting in approximately 19- and 12-fold-selectivity. However, Uddin et al. [116] found the affinity to COX-1 to be much higher at >4 µM, which means that the radioligands lack the required selectivity and hence cannot be used for the in vivo imaging of COX-2.

As usual, the sulfonamide derivative showed a higher retention in the blood than the methylsulfone derivative at all time points. The brain concentration of both tracers in healthy rats was moderate [117] at 10 min p.i. 0.25 and 0.23 %ID/g for [^125^I]IMTP and [^125^I]IATP respectively; at 180 min, 40% of the methylsulfone derivative [^125^I]IMTP remained in the brain while the brain concentration of the other remained almost unchanged at the same time point. The stability of the tracers was demonstrated by their low uptake in the thyroid and the stomach [113].

In addition to the celecoxib derivatives, Kuge et al. [118] developed a lumiracoxib derivative, [^125^I]FIMA (Figure 5), with the chlorine substituent in the lumiracoxib molecule substituted for iodine. This resulted in decreased affinity to COX-2 from 0.8 µM for lumiracoxib to 2.5 µM in the same assay, which might be due to the relatively larger size of the iodine substituent, which was too big for the COX-2 binding pocket, as pointed out by Tietz et al. [90]. Notwithstanding, it also showed tracer uptake in LPS/IFN-γ-stimulated macrophages in comparison to the non-stimulated ones. In biodistribution experiments in male SD rats, it showed no in vivo deiodination, indicated by the low uptake in the thyroid and stomach. There was, however, low uptake of the tracer in the brain, with 0.04 %ID/g at 10 min p.i. This could be due to the presence of a carboxylic acid group in the molecule, which although it decreased the lipophilicity of the tracer may ionize in vivo. 

More recently, Tietz et al. [90] reported a tracer [^125^I]pyricoxib, a radio-iodinated analog of the [^18^F]fluorine-labeled pyricoxib, in the hope that it will retain the same affinity to COX-2 and selectivity over COX-1 as tracer [^18^F]pyricoxib. This however was not the case as [^125^I]pyricoxib displayed approximately twelve times less inhibitory potency than [^18^F]pyricoxib, and around five times less than the ligand with a bromine atom in the 4-benzyl position. The steric bulk of iodine was believed to have played a role in this loss of potency, although it displayed a better IC_50_ than the other aforementioned radioiodinated COX-2 inhibitors, with improved selectivity over COX-1 as well. The group plans to radioiodinate the ligand with ^124^I in order to use the ligand for PET imaging as well as SPECT imaging [90].

**Figure 5 ijms-24-17419-f005:**
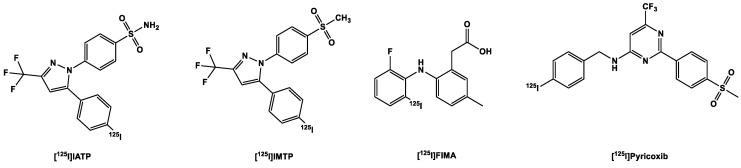
Structure of radioiodinated COX-2 tracers.

#### 3.3.2. [^99m^Tc]Technetium-Labeled COX-2 Tracers

In 2004, Yang et al. [119] reported a ^99m^Tc-labeled celecoxib (Figure 6) with its sulfonamide group modified by reacting it with ethyl 2-isocyanatoacetate to obtain an intermediate (an ester form of CBX). In a subsequent aminolysis reaction with ethylenediamine, EA-CBX was synthesized. These modifications not only enabled the attachment of the chelator but also allowed for sufficient distance between the pharmacologically active part of the tracer and the chelator. In the final step, EC–CBX was synthesized in a condensation reaction of EA-CBX with L,L-ethylenedicysteine (EC). The precursor EC–CBX was then labeled as [^99m^Tc]pertechnetate.

**Figure 6 ijms-24-17419-f006:**
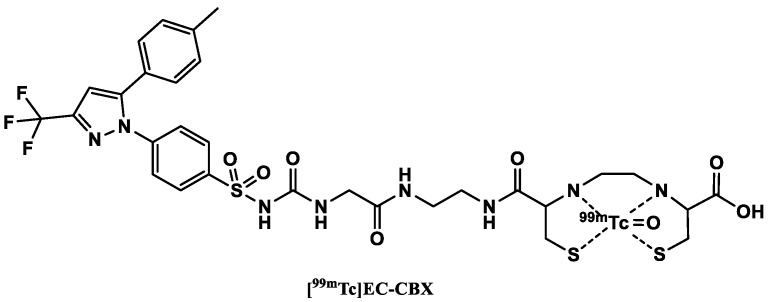
Structure of ^99m^Tc-labeled COX-2 tracers.

An in vitro cellular tracer uptake assay in three cancer cell lines surprisingly showed a significant increase in tracer uptake after pretreatment with celecoxib. The authors suggested that this might be due to the induction of COX-2 expression by celecoxib. A more plausible explanation was suggested by Tietz et al. [78] who believed that the increased uptake was due to inhibition of P-glycoprotein by celecoxib in parallel to blocking COX-2-mediated uptake of the tracer.

Biodistribution experiments in breast tumor bearing rats showed that the uptake of the tracer was higher in tumors compared to the administration of only the ^99m^Tc-labeled chelator ([^99m^Tc]EC). The tracer additionally displayed favorable tumor–muscle ratios as a function of time (tumor–muscle ratio of 4.3 at 30 min and 9.7 at 4 h). The tumor/blood ratios were lower than one due to high blood retention of the tracer. Planar scintigraphy was also performed in tumor-inoculated nude mice, rats and rabbits. Planar images confirmed that the visualization of the tumors could be carried out from 30 min–4 h p.i. in the animal models.

Based on their results, the authors believe that the tracer could be used for the assessment of COX-2 expression of cancer. EC–CBX may also be chelated with other isotopes as (^68^Ga, ^61^Cu) for PET imaging and ^188^Re for internal radionuclide therapy [119]. Since the sulfonamide functional group is believed to be essential for binding to the COX-2 enzyme, attaching a chelating agent directly to this group might diminish the affinity of the tracer to the target [120].

Some years later, Farouk et al. [121] presented another ^99m^Tc-labeled celecoxib. Celecoxib in this case was labeled without a chelator, but in optimized conditions and, notably, at room temperature. The structure of the final complex formed, however, was not revealed by the authors.

Biodistribution studies carried out in healthy mice showed a gradual decrease in the blood concentration of the tracer, with up to 47% of the tracer concentration at 15 min p.i. remaining at 4 h p.i. The kidneys also showed high tracer uptake, an indication of renal excretion. Biodistribution experiments carried out in sterile inflamed mice showed nearly the same biodistribution results of the tracer as in healthy mice. Nonetheless, the inflamed muscle showed higher tracer uptake 1.7, 2.1 and 2.5 times higher at 15 min, 1 h and 4 h, respectively, than the non-inflamed tissue. The authors suspected that this higher uptake might be due to increased vascularization of the inflamed muscle, which facilitated a higher supply of the tracer to the area of interest. No additional blocking experiments were performed to determine the specificity of the uptake.

Chadha et al. [122] carried out further experiments with the same tracer as Farouk et al. [121], which they radiolabeled using a different approach. They discovered a higher uptake of the tracer in the intestines of the 1,2-dimethylhydrazine (DMH)-treated rats compared to healthy control rats. A significant uptake (2, 2.9 and 2.4 times higher at 0.5 h, 2 h and 4 h p.i., respectively) was also observed in the tumor site relative to the site adjacent to the tumor at all time points. Planar SPECT images at 4 h p.i. showed significant uptake of the tracer in the tumor site, confirmed with histopathological studies, which suggest that the tracer has better selectivity towards neoplastic tissue in the colon of the DMH-treated rats in comparison to control rats.

In conclusion, considerable progress has been made so far in the development of COX-2 tracers that can be used for the in vivo imaging of the COX-2 enzymes, considering the obstacles met at the beginning, which included low metabolic stability, high blood retention of the sulfonamide analogs due to off-target binding to CA, and off-target binding to p-glycoprotein, which is a nuisance as it decreases the tumor–muscle ratio, and suitability of biological models for the in vivo evaluation of the prospective tracers. 

As already discussed for the TSPO 18 kDa protein, COX-2 expression is upregulated during pathology, both in acute and chronic inflammatory conditions. However, neuroinflammatory responses to pathological stimuli differs: the upregulation of COX-2 occurs more quickly but is more transient, with a return to baseline in a matter of hours [80]; in contrast, TSPO upregulation can take several days and may last for weeks [123]. Moreover, the location of COX-2 after induction is largely restricted to neurons and brain endothelial cells [76,124], whereas TSPO is minimally upregulated in neurons but highly in microglia and astroglia. This transient upregulation of COX-2 expression in neuroinflammatory responses might allow for the usage of COX-2 tracers for the measurement of acute as well as chronic inflammation. Conversely, TSPO tracers will be useful to image chronic inflammatory conditions, as might be found in NDDs [80,125]. In order to image the transient upregulation of COX-2, it is important that the prospective tracer have fast brain pharmacokinetics, which include not only the ability to quickly cross the BBB into the brain but also cross the lipid bilayer of the cell organelles in which the enzyme is located [86,87,88], as well as possess specific binding with a fast clearance of nonspecifically bound tracers from the brain [126]. This could exclude iodine-labeled tracers, which due to their relatively larger size of radionuclide may result in reduced affinity to COX-2 [90,113] and decreased brain uptake of the tracers [110,113]. However, compared to fluorine-18-labeled analogs, iodine-124/125-labeled analogs may show fewer off-target bindings, as has likely been the case for [^18^F]SC58125 (Figure 4) [110] compared to [^125^I]IMTP (Figure 5) [113].

Although this is only a minor problem, there are still concerns about selectivity in COX-1. Recently, the methylsulfone tracers [^11^C]MC1 [80,84], [^18^F]pyricoxib [78,109] and [^125^I]pyricoxib [90] and the celecoxib analog [^18^F]6 [87] have been developed, and they showed minimal off-target binding to CA. Nevertheless, there is still a need for an improved and validated biological model, which will serve as the gold standard for the evaluation of new COX-2 tracers and enable crossover between studies. A list of the reported COX-2 radiotracers is presented in Table 2.

## 4. PET and SPECT Imaging of TSPO and COX-2 in Non-Neuroinflammatory Diseases

### 4.1. Introduction

Inflammation is significantly involved in a wide variety of diseases, which are not limited only to neurodegenerative pathologies. As a matter of fact, chronic (sub-clinical) inflammation is believed to be the most significant cause of death in the world, with not less than 50% associated with inflammation-related diseases such as cancer, stroke, ischemic heart disease, chronic kidney disease, diabetes mellitus, non-alcoholic fatty liver disease and the aforementioned NDD [127,128]. 

### 4.2. Pulmonary Inflammation

The CNS communicates with other organs, such as the skin [129], intestines, kidneys and lungs [130,131,132,133]. As such, the activities of the microbiomes in these organs could in turn affect brain activities, especially in relation to disease development [127,129,130]. The lungs, for instance, communicate directly with the external environment, and therefore, it is not surprising that emerging evidence shows that CNS diseases are related to air pollution, which increases the risk of AD, PD, autism and cognitive dysfunction in the elderly and correlates with higher incidence of stroke. Albeit not directly, air pollution may activate microglia in due course induce or accelerate inflammation in the brain via immune, endocrine or other mechanisms [133]. 

Researchers at the Institute for Neuroimmunology and Multiple Sclerosis Research at University of Göttingen have established that changes in the lung microbiome plays a role in the susceptibility of the brain to autoimmune inflammation such as multiple sclerosis [132]. The neuropathology of NDDs such as AD and PD are also linked to pulmonary bacteria [130]. Moreover, clinical studies have showed that chronic obstructive pulmonary disease (COPD) is linked to cerebrovascular disease by an increment in white matter lesions and predisposes patients to ischemic as well as hemorrhagic stroke [133,134]. Chronic lung pathologies or prolonged exposure to air pollutants could facilitate brain inflammation. Recent experimental evidence suggests that there are several lung-to-brain pathways involved in signaling to the brain during acute lung inflammation induced by the intratracheal injection of LPS in mice, which include circulating cytokines and immune cells [135].

Biomarkers of inflammation such as TSPO are upregulated in activated macrophages in COPD and asthma, as well as malaria-associated acute respiratory distress syndrome (MA-ARDS). Although specific studies investigating the use of TSPO and COX-2 imaging in lung inflammation are limited, it is conceivable that these imaging modalities could be explored in the context of pulmonary inflammatory conditions, such as pneumonia, bronchitis, COPD and other respiratory disorders characterized by inflammation. PET-based TPSO radiotracers could aid the visualization and the study of lung inflammation in order to better understand the mechanism of lung inflammation and lung-to-brain communication [133]. 

Goggi et al. [136] were able to longitudinally image TSPO in the lungs of rodent models of MA-ARDS using the second-generation TSPO tracer [^18^F]FEPPA (Figure 1). [^18^F]FEPPA was able to track changes in pulmonary accretion of interstitial inflammatory macrophages and MHC-II-positive alveolar macrophages in *Plasmodium berghei ANKA*-infected mice in correlation with parasitemia. 

In a study by Chen et al. [137], using another second-generation TSPO tracer ([^11^C]PBR28) (Figure 1), the authors demonstrated that [^11^C]PBR28 can better distinguish macrophage-dominant from neutrophilic-dominant inflammation than [^18^F]FDG in mouse models of lung inflammation. Moreover, they showed that [^11^C]PBR28 can quantify M2-polarized TSPO-expressing-macrophages in the lungs. There was around a 3-fold increase in this macrophage phenotype 49 days post-Sendai virus (SeV) infection compared to the monocyte-deficient mouse model with attenuated chronic inflammatory response also on day 49 post-infection. In addition, minimal recruitment of macrophages was observed in WT mice on the third day post-infection. Still, no change in [^11^C]PBR28 uptake was observed 24 h after the intranasal instillation of endotoxin (LPS) in WT mice. This was well in line with the fact [138] that an M1 macrophage-driven inflammatory response predominates with an abundance of neutrophils at this stage of the infection. Finally, they were able to confirm the recruitment of macrophage (TSPO) to mouse lungs using immunostaining [137]. This was in agreement with a similar experiment using human lung sections [137]. In human lung sections, the intensity of TSPO staining was significantly higher in CD68+ cells compared to neutrophils. These experiments revealed that [^11^C]PBR28 uptake not only depends on macrophage recruitment but may also depend on the macrophage phenotype. 

Contrary to the study by Chen et al. [137], a previous study by Hatori et al. [139] was able to demonstrate that the second-generation tracer [^18^F]FEDAC (Figure 1) can detect TSPO expressed in macrophages and neutrophils in the lungs of mice instilled with LPS. Considering the duration of the experiment, it is apparent that only M1-polarized macrophages and neutrophils predominated and hence were the main source of the observed signals. There was a significant difference in the uptake of [^18^F]FEDAC in this mouse model of LPS-induced lung inflammation compared to the uptake of [^11^C](R)-PK11195 24h after LPS instillation [139]. Western blot assay confirmed an increased TSPO expression over time (2, 6 and 24 h) after LPS instillation compared to controls. IHC analysis further substantiated that TSPO was expressed by activated neutrophils and macrophages after pulmonary LPS instillation, with time-dependent recruitment of macrophages over a span of 24 h. However, it remains to be determined whether [^18^F]FEDAC may also label M2-polarized macrophages such as [^11^C]PBR28 in cases of chronic inflammation or whether its efficacy is only limited to acute pulmonary inflammation. 

The reported study by Chen et al. [137] corroborated a study by Jones et al. [140] conducted nearly twenty years prior to that of Chen and his colleagues. In their study in rabbits, Jones et al. [140] monitored the kinetics of lung macrophages after particle challenge to the lung using the first-generation TSPO tracer [^11^C]R-PK11195. In summary, this study also showed that TSPO tracers can be applied in the assessment of effects of airborne pollutants. 

Using [^11^C]DPA-713, Shah et al. [141] sought to unearth the mechanism of organ-level pathophysiology that ensues during Ebola virus infections. Longitudinal PET imaging together with various disease biomarkers and IHC indicated that, compared to baseline values, there is sustained loss of TSPO in lungs and spleen owing to the local depletion of monocytes, macrophages and dendritic cells. This is accompanied by a reactive haematopoietic activation in the bone marrow. Similar approach using TSPO PET ligands could help to noninvasively assess the pathogenesis and progression of other systemic inflammatory and infectious diseases in real time. 

The incorporation of longer-lived iodine radioisotopes in DPA-713 has further facilitated the study of other infectious pulmonary diseases such as tuberculosis (TB) as well as SARS-CoV-2 infection. Ordonez et al. [142] used the iodine-125 (t_1/2_ 59.5 days) radiolabeled [^125^I]DPA-713 to monitor the potency of a novel bedaquiline-containing TB drug regimen. This regimen is planned for patients with multidrug-resistant (MDR) tuberculosis [142].

TB remains still an important cause of human death amongst curable diseases and its complete elimination by 2050 is the target of the World Health Organization (WHO). TB-associated inflammation is characterized by the presence of activated macrophages, which TSPO tracers such as [^125^I]DPA-713 could image and consequently furnish invaluable real-time information about the outcome of TB treatment and the possibility of relapses [142]. Pulmonary SPECT with [^125^I]DPA-713 established significant correlation with the bactericidal potency of TB treatments both standard and bedaquiline-containing compared to [^18^F]FDG PET in a C3HeB/FeJ TB model. Moreover, a time-dependent decrease in the cytokine levels interferon-γ (IFN-γ) and tumor necrosis factor-α (TNF-α) was detected with tuberculosis treatments, which correlated with [^125^I]DPA-713 uptake. Being a low-energy gamma emitter, iodine-125 was exchanged with the higher energy emitting iodine-124, which has better tissue penetration. [^124^I]DPA-713 PET showed up to 4-fold higher uptake in the infected tuberculosis lesions than uninfected controls [142]. 

Ruiz-Bedoya et al. [143] also used [^124^I]DPA-713 to evaluate the immune response in a hamster model of SARS-CoV-2 infection. PET/CT scans established that the tracer was trapped by activated macrophage cells in pulmonary lesions. Additionally, there were significant gendered differences observed regarding [^124^I]DPA-713 PET/CT in the hamsters: female hamsters displayed higher PET activity compared to the males. These findings were confirmed via optical imaging, immunofluorescence and flow cytometry. 

### 4.3. Autoimmune Diseases

Rheumatoid arthritis (RA) a heterogenic immunopathology characterized by chronic synovial inflammation (resulting oftentimes in structural joint damage) can also be diagnosed by non-invasive PET tracers of inflammation [144,145]. By this route, quantitative measurements of changes in perfusion can be conducted in order to monitor and predict the efficacy of disease-modifying anti-rheumatic drugs. This is necessary since some RA patients fail to respond to therapy [146,147]. Histopathological analysis of synovial tissue biopsies, the classical method to evaluate treatment efficacy and immunopathological features of RA, is invasive [138]. The samples are acquired via ultrasound-guided biopsies or arthroscopy. Moreover, this classical method is delimited by the number of joints that can be examined [144,148]. 

The first-generation TSPO tracer [^11^C]R-PK11195 (Figure 1) was able to image upregulated TSPO in activated macrophages in RA. Van der Krogt et al. [144] were able to (significantly) quantitatively visualize synovitis in severely inflamed joints and joints with mild/moderate signs of inflammation. Interestingly, [^11^C]R-PK11195 was also able to visualize subclinical disease activity in an uninflamed knee contralateral to an inflamed knee. PBR staining of the sublining of synovial tissue correlated significantly with the uptake of [^11^C]R-PK11195 in the joints as well as CD68 staining of macrophages. However, macrophage specificity of the CD68 staining was later disproved by Narayan et al. [149]. Gent et al. [150] also corroborated the usefulness of [^11^C]R-PK11195 for the detection of subclinical RA. Using the same tracer, they demonstrated that [^11^C]R-PK11195 can visualize subclinical RA in patients who expressed anti-citrullinated protein antibodies (ACPAs). A two-year follow-up confirmed the previous discovery. Gent et al. further demonstrated that [^11^C]R-PK11195 PET scans can predict flares in RA patients [151] even better than MRI scans [152]. These were patients without clinical arthritis during or after treatment. Nevertheless, being a relatively lipophilic tracer (ClogP 4.6), its background uptake in periarticular tissues was relatively high and, thus, hindered the visualization of more subtle synovitis.

Narayan et al. [149] subsequently was able to detect inflammation in the tibiofemoral joints of RA patients with the less lipophilic tracer [^11^C]PBR28 (ClogP 3.4) with a significantly higher signal than in healthy controls. These results were confirmed by both synovial tissue autoradiography (with [^3^H]PBR28) and IHC staining. Interestingly, immunofluorescence showed that in addition to the macrophages, there are other TSPO-expressing cells in the RA pannus, namely CD4^+^-T lymphocytes and fibroblast-like synoviocytes (FLS) (activated stromal cells). The in vitro assessment of TSPO mRNA expression and [^3^H]PBR28 binding revealed that the highest expression of TSPO was seen in activated FLS cells, nonactivated M0 macrophages, and activated M2 reparative macrophages. Low expression of TSPO was observed in M1 macrophages, monocytes, unstimulated FLS cells and, with the overall lowest expression, in activated and nonactivated CD4^+^ T cells. This remarkable contribution of activated FLS cells to TSPO PET signals from the RA pannus would be beneficial in the determination of responses to FLS cells-targeted therapies.

Bruijnen et al. [148] studied the relatively less lipophilic O-[^18^F]fluoroethylated DPA-TSPO tracer [^18^F]DPA-714 (Figure 2) (ClogP 3.3) and its O-[^11^C]methylated analog [^11^C]DPA-713 (ClogP 3.1) in RA patients. [^18^F]DPA-714 PET showed only a marginal improvement over [^11^C]R-PK11195. Even with a slightly lower background uptake of [^18^F]DPA-714 (Figure 2), the target-to-background (T/B) ratio of both tracers did not differ significantly. The mean SUV values of PET-positive joints did not differ between the two tracers. A combination of lower background uptake of [^11^C]DPA-713 and higher absolute uptake in the inflamed joints resulted in up to a two-fold higher T/B ratio than that of the two other tracers. The outperformance of [^18^F]DPA-714 by [^11^C]DPA-713 may be related to the lower lipophilicity of the latter. Both DPA tracers showed no susceptibility to SNP polymorphism, unlike in brain imaging, which is indicative of the possible absence of the effect TSPO polymorphism in RA imaging. 

Interestingly, it has also been reported that COX-2 can serve as a biomarker for RA PET imaging. Shrestha et al. [80], using the PET tracer [^11^C]MC1 (Figure 3), demonstrated that there was significantly higher uptake of the tracer in inflamed joints, consistent with symptoms of the patients with RA. A PET scan with the third-generation TSPO tracer [^11^C]ER176 (Figure 2) was also carried out, which showed results similar to those found with [^11^C]MC1. The specificity of [^11^C]MC1 uptake was determined in another scan after a per os administration of 400 mg of celecoxib. Celecoxib showed no effect on the uptake of the [^11^C]ER176 but decreased the uptake of [^11^C]MC1 in the joints with a more consistent blocking of brain uptake of the tracer.

The TSPO biomarker could also be useful for the in vivo imaging of inflammation in large blood vessels, e.g., in the aorta and its main branches and other large vessel vasculitis (LVV) [153]. LVV is a characterized by a dysfunctional immune reaction to injury that encourages intramural vascularization, intimal hyperplasia and adventitial thickening, leading to loss of vessel integrity and ischemic damage of dependent organs [153,154]. LVV is marked by a local intramural chronic granulomatous inflammation of the aortal vessel wall and that of its main branches, which is characterized by the presence of macrophages that highly express TSPO [154,155]. 

Molecular imaging with PET imaging agents enables the diagnosis of vasculitis, especially in vascular beds for which biopsy is not possible [156]. Supported by contrast-enhanced computer tomography angiography (CTA), hybrid PET scanners could enable a clear delineation of the blood vessel wall anatomy with which the tracer signals could be co-registered [156,157]. 

[^11^C]R-PK11195 (Figure 1) in combination with CTA has also been used for the imaging of LVV in patients with systemic inflammatory disorders [156,157]. A higher vascular uptake of the tracer was seen in symptomatic patients compared to asymptomatic patients [157]. This study was corroborated by another study by the same group [156]. Nonetheless, the imaging of LVV is limited by the proximity of the inflammation site to the blood pool and insufficient thickness of the arterial wall, which leads to partial and spillover effects which do not differ between symptomatic and non-symptomatic patients. This calls for further quantitative studies in addition to the already-implemented spillover correction and quantification of receptor kinetics [155,156,157].

### 4.4. Cardiovascular Pathology

TSPO upregulation has been observed in cardiac diseases such as myocarditis [146], arrhythmia [158], atherosclerosis [159] and cardiac hypertrophy [160]. In cardiovascular diseases such as myocardial infarction (MI) [161] and artrial arrhythmia [158], TSPO could play the role of more than a diagnostic biomarker, as it can also serve as a therapeutic target. For this reason, the quantification of TSPO expression via TSPO tracers might provide invaluable information, which would help to individualize patient treatment. 

Mou et al. [162] employed [^18^F]F-DPA (Figure 1) from the DPA family (Figure 3) to evaluate cardiac inflammation in rats post MI. [^18^F]F-DPA displayed high stable cardiac uptake with fast clearance from nearby organs such as the lungs, which enabled improved heart imaging. There was a higher normalized SUV ratio (NSR) of [^18^F]F-DPA to [^13^N]NH_3_ in the infarct and peri-infarct regions compared to remote regions. Blocking studied with PK11195 confirmed the specificity of the tracer for TSPO, as also did H&E staining. NSR evaluation using [^13^N]NH_3_ as a reference was absolutely necessary, since TSPO is also highly expressed in healthy cardiomyocytes and, therefore, may confound the results of cardiac inflammation. Furthermore, the uptake of [^18^F]F-DPA was lower in the infarct region compared to the remote region as a result of mitochondrial dysfunction in the former, which negatively affected visualization [162]. Overall, higher uptake in the remote region compared to the infarct region suggests that cardiac TSPO imaging is a due to mitochondrial dysfunction in dying cardiomyocytes and activated inflammatory cells [163]. 

Using a polar map of [^99m^Tc]sestamibi as a reference, Thackeray et al. [163,164] analyzed high signals of [^18^F]flutriciclamide ([^18^F]GE180) (Figure 2) in infarcted regions in mice. [^18^F]GE180 was able to image infiltrating CD68^+^ macrophages in the acutely infarcted regions at one week post-MI. It was also able to autonomously predict contractile dysfunction at eight weeks post-MI, during which TSPO signals were associated with mitochondrial dysfunction in failing but non-infarcted cardiomyocytes remote from the infarcted regions. They also assessed the influence of MI on both cardiac and brain inflammation using [^18^F]GE180 (Figure 2). Neuroinflammation is believed to correspond to cardiac dysfunctions in the early phase of post-infarct myocardial inflammation and in the late phase of chronic heart failure. This effect could be attributed to increased proinflammatory cytokines and diminished cerebral blood flow as well as increasing levels of angiotensin II [164]. Moreover, elevated cerebral TNF-α and activated microglia were observed in mice with congestive heart failure, which was accompanied by cognitive impairment [165]. 

Macrophages also play an important role in the development of atherosclerosis [166], which is a systemic condition with increased risk of stroke or myocardial infarction. It comprises inflamed localized plaques with an abundance of activated macrophages with upregulated TSPO expression [167] and somatostatin type-2 receptors (SSTR2). In addition to the former, the latter has proven to be another useful target for the imaging of inflammation with such tracers based on the somatostatin analog octreotide as [^68^Ga]DOTATOC [166], [^68^Ga]DOTATATE [168], and [^64^Cu]DOTATATE [169]. Owing to the longer half-life of copper-64 (12.7 h) compared to gallium-68 (67.7 min), imaging could be carried out a later time point, which allows time for the clearance of both the unspecifically bound tracer and its clearance from healthy TSPO-expressing tissue hence improving spatial resolution. Moreover, in a head-to-head comparison with [^68^Ga]DOTATOC, it was seen that, compared to the uptake of [^64^Cu]DOTATATE, the uptake of [^68^Ga]DOTATOC was significantly lower in vascular regions and did not correlate with cardiovascular risk factors [167].

Taking advantage of another long-lived radio-isotope, iodine-125 (59.5 days), Foss et al. [170] were able carry out twenty-four-hour delayed SPECT imaging with [^125^I]DPA-713. By doing so, the tracer was given enough time to clear from healthy TSPO-expressing tissues such as the lungs and the heart, and from the liver as well, thus enabling the imaging of inflamed plaques containing CD68-expressing phagocytes with low background tracer signal with high contrast in ApoE−/− mice regardless of diet. SPECT images revealed focal uptake of [^125^I]DPA-713 at the aortic root along the descending aorta and within the myocardium of all the ApoE−/− mice compared to age-matched controls [170].

Kopecky et al. [171] were able to show via PET imaging that there was around a 3-fold uptake of [^18^F]PBR111 (Figure 1) in ApoE−/− mice on a high-fat diet (3 or 12 weeks) with mature atherosclerosis compared to controls. They were able to show that TSPO colocalized with the macrophages which infiltrated the atherosclerotic plaques, but not with tissue-resident macrophages [171].

Gaemperli et al. [172] also demonstrated the prospective use of [^11^C]PK11195 (Figure 1) PET to non-invasively detect and quantify intraplaque inflammation in patients with carotid stenosis. Their findings showed that with [^11^C]PK11195 PET supported by CTA, it is feasible to differentiate between recently symptomatic and asymptomatic plaques. Ipsilateral plaques with lower CT attenuation and heightened [^11^C]PK11195 uptake was observed in patients with a recent ischemic event [172].

## 5. Conclusions

In this review, we highlighted several interesting TSPO and COX-2 PET and SPECT tracers. The background of the development of some of the tracers was provided in order to highlight some of the pitfalls that should be avoided in the development of future tracers. Apart from the biomarkers of inflammation mentioned in this review, there exist other targets, which may be more suitable for the investigation of specific pathologies. Nevertheless, TSPO and COX-2 are the most studied biomarkers of inflammation, which explains the high number of tracers already developed for these targets. Some of these tracers can be labeled with both radioisotopes for PET and SPECT imaging. Radiolabeling with SPECT radio-isotopes enables us to monitor the pharmacokinetics of tracers with slow clearance from the organs of interest. This facilitates preclinical assessments, since imaging can be performed for hours post-administration. In particular, the positron-emitting iodine-124 and copper-64 are useful in this regard. Additionally, there is also a potential to translate the use of such labeled tracers to clinical settings. 

Recent advancements in TSPO ligands have provided improved imaging options, but their clinical application can be limited by genetic variations or SNPs in the TSPO gene. Further research is needed to understand the binding requirements of WT and variant TSPO, which will aid in the development of more effective and specific TSPO ligands for clinical applications. Inflammatory responses should be actively halted when their detrimental effects outweigh their benefits in order to avoid unnecessary bystander damage to tissues. Fortunately, some of the aforementioned TSPO tracers, such as [^11^C]R-PK11195 (in RA cases), are able to detect inflammation even before clinical presentation of symptoms. Applications of tracers targeting COX-2 may be limited due to its transient expression and consequent rapid return to baseline. Such pitfalls may present a challenge in accurately detecting and monitoring chronic neuroinflammation. Hence, the success of COX-2 targeted imaging may depend on the judgment and expertise of the clinician in each individual case: careful assessment and interpretation of imaging results in order to determine its clinical relevance. Moreover, further research and development of alternative tracers with improved characteristics are needed to overcome these limitations and enhance the effectiveness of neuroinflammation imaging to track inflammation. This might in turn facilitate the development of drugs for the treatment of both infectious and non-infectious diseases such as cancer and autoimmune diseases, as well as neuroinflammation and associated neurodegeneration. 

## Figures and Tables

**Figure 1 ijms-24-17419-f001:**
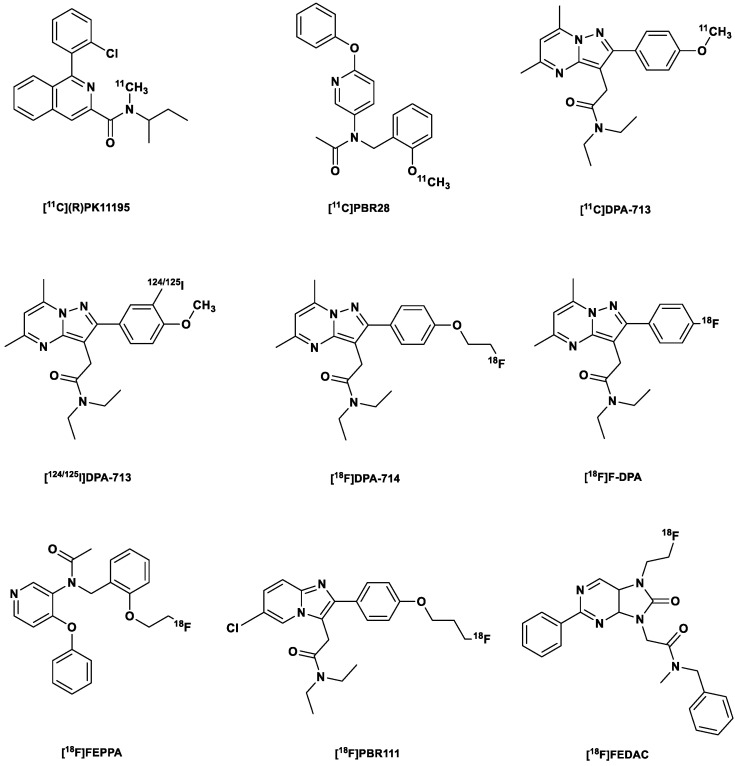
Structures of [^11^C](R)PK11195) and second-generation TSPO tracers.

**Figure 2 ijms-24-17419-f002:**
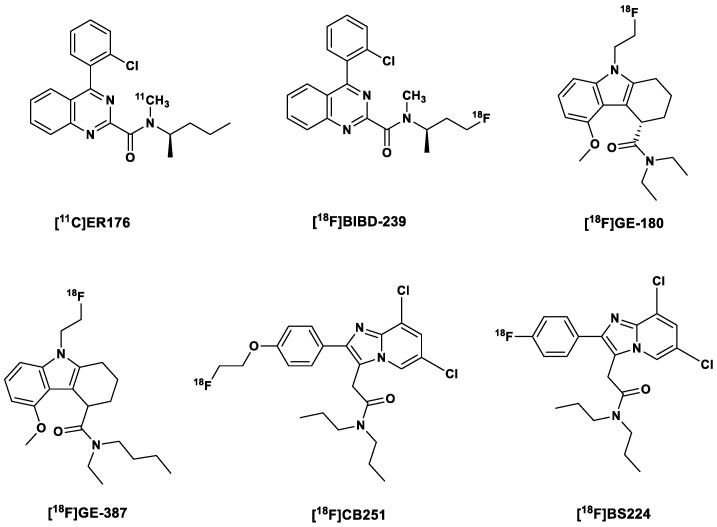
Structures of third-generation TSPO tracers.

**Table 1 ijms-24-17419-t001:** List of the reported TSPO radiotracers.

№	Generation	Radiotracer
1.	First	[^11^C](R)PK11195
2.	Second	[^11^C]PBR28
3.		[^11^C]DPA-713
4.		[^123^I]/[^124^I]/[^125^I]DPA-713
5.		[^18^F]DPA-714 (PBR099)
6.		[^18^F]F-DPA
7.		[^18^F]FEPPA
8.		[^18^F]PBR111
9.		[^18^F]FEDAC
10.	Third	[^11^C]ER176
11.		[^18^F]BIBD-239
12.		[^18^F]GE-180 (Flutriciclamide)
13.		[^18^F]GE-387
14.		[^18^F]CB251
15.		[^18^F]BS224

**Table 2 ijms-24-17419-t002:** List of reported COX-2 tracers.

№	Imaging Modality	Radiotracer
1.	PET	[^11^C]Celecoxib
2.		[^11^C]Rofexcoxib
3.		[^11^C]MC1
4.		[^18^F]1 (Celecoxib derivative)
5.		[^18^F]2 (Celecoxib derivative)
6.		[^18^F]3 (Valdecoxib derivative)
7.		[^18^F]4 (Celecoxib derivative)
8.		[^18^F]Pyricoxib
9.	SPECT	[^125^I]IATP
10.		[^125^I]IMTP
11.		[^125^I]FIMA (Lumiracoxib derivative)
12.		[^125^I]Pyricoxib
13.		[^99m^Tc]Celecoxib

## Data Availability

Not applicable.

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
