# Peer review of "Radiotracers for Imaging of Inflammatory Biomarkers TSPO and COX-2 in the Brain and in the Periphery"

_ijms, 2023, doi:10.3390/ijms242417419_

Round 1

Reviewer 1 Report

Comments and Suggestions for Authors

Manuscript ID: ijms-2707481

Review 1

Radiotracers for Imaging of Inflammatory Biomarkers TSPO 2 and COX-2 in the Brain and in the Periphery

By Uzuegbunam et al

Neuroinflammation plays a central role in various neurodegenerative conditions. PET scans to monitor neuroinflammation offers a means of diagnosing and tracking disease progression, as well as evaluating the effectiveness of interventions. This review is therefore dealing with the topic of significance that is relevant to may clinical presentations including neuro degenerative diseases, brain, and spine. In the foreseeable future, probably the diagnosis and monitoring of neurodegenerative diseases will involve the use of specialized imaging agents either through PET or SPECT scanners.

Although the review is well presented barring to some typos, following are some observations from reviewer’s point of view:

The authors need to justify their choice of Translocator protein (TSPO) and cyclooxygenase-2 (COX2) to consider them together. While PET radioligands targeting TSPO have shown promise in detecting neuroinflammation TSPO, there are other specific choices like Dopamine transporters (DAT) and Vesicular monoamine transporter (VMAT)too.

Recent advancements in the field have introduced second and third generation TSPO ligands, providing less selective imaging options with respect to the TSPO variants. Although even third generation tracers are also sensitive to some of the specific genetic variation/SNPs in target limiting their clinical applicability. It will therefore be vital to gain a greater understanding of how the binding requirements of WT and variant TSPO differ.  There may be small structural difference between different mutants and SNPs that will show discrimination in ligand binding and therefore may show differences in scan. “Genotypic variations” affecting binding affinity differences are slowly appearing in the literature. Some of the authors also have expressed requirements for genotyping before scanning to exclude lower affinity binding for ligands. However, this vital point has not been adequately addressed in this review.

Regarding COX 2 expression, the authors noted that COX2 expression is transient (that return to baseline within few hours) and restricted for neurons and endothelial cells, that makes only a narrow window available for investigation. Therefore, success of the scan depends on the judgement of the clinician in each case. These features may limit COX 2 to be practically useful marker for chronic neuro-inflammation. The authors must highlight limitations of these tracers targeting these ligands.

Comments on the Quality of English Language

Some typos noticed in the text

Author Response

Thank you to the reviewers for their valuable comments and suggestions on our manuscript. We sincerely appreciate the time and effort they have dedicated to providing feedback. In response to their critiques and suggestions, we have carefully revised the manuscript and have prepared a detailed point-by-point response to address each comment. We believe that these revisions have significantly strengthened the manuscript and have addressed all the concerns raised by the reviewers.

Please find attached our point-by-point response to the reviewers' comments.

Reviewer 2 Report

Comments and Suggestions for Authors

The publication titled "Radiotracers for Imaging of Inflammatory Biomarkers TSPO and COX-2 in the Brain and in the Periphery" is intriguing and could serve as a significant and robust source for further research on radiotracers crucial in various inflammatory diseases. However, the extensive text makes it challenging to extract the data. To enhance the publication's reception among scientists, the authors might consider adding tables or graphs to summarize the most important data from the manuscript. Additionally, including a list of abbreviations would improve the manuscript. There are no further comments. I believe the work is of high standard, and the authors have thoroughly explored the topic.

Author Response

Thank you to the reviewers for their valuable comments and suggestions on our manuscript. We sincerely appreciate the time and effort they have dedicated to providing feedback. In response to their critiques and suggestions, we have carefully revised the manuscript and have prepared a detailed point-by-point response to address each comment. We believe that these revisions have significantly strengthened the manuscript and have addressed all the concerns raised by the reviewers.

Please find attached our point-by-point response to the reviewers' comments.

Reviewer 2: Neuroinflammation plays a central role in various neurodegenerative conditions. PET scans to monitor neuroinflammation offers a means of diagnosing and tracking disease progression, as well as evaluating the effectiveness of interventions. This review is therefore dealing with the topic of significance that is relevant to may clinical presentations including neuro degenerative diseases, brain, and spine. In the foreseeable future, probably the diagnosis and monitoring of neurodegenerative diseases will involve the use of specialized imaging agents either through PET or SPECT scanners.

  • Tables summarizing the reported tracers are included on pages 9 and 18
  • A table of Abbreviation and acronyms is added on page 24
